# Improved A* Algorithm for Path Planning of Spherical Robot Considering Energy Consumption

**DOI:** 10.3390/s23167115

**Published:** 2023-08-11

**Authors:** Hao Ge, Zhanfeng Ying, Zhihua Chen, Wei Zu, Chunzheng Liu, Yicong Jin

**Affiliations:** 1National Key Laboratory of Transient Physics, Nanjing University of Science & Technology, Nanjing 210094, China; gehao@njust.edu.cn (H.G.);; 2School of Energy and Power Engineering, Nanjing University of Science & Technology, Nanjing 210094, China

**Keywords:** path planning, spherical robot, improved A-star algorithm, energy consumption

## Abstract

Spherical robots have fully wrapped shells, which enables them to walk well on complex terrains, such as swamps, grasslands and deserts. At present, path planning algorithms for spherical robots mainly focus on finding the shortest path between the initial position and the target position. In this paper, an improved A* algorithm considering energy consumption is proposed for the path planning of spherical robots. The optimization objective of this algorithm is to minimize both the energy consumption and path length of a spherical robot. A heuristic function constructed with the energy consumption estimation model (ECEM) and the distance estimation model (DEM) is used to determine the path cost of the A* algorithm. ECEM and DCM are established based on the force analysis of the spherical robot and the improved Euclidean distance of the grid map, respectively. The effectiveness of the proposed algorithm is verified by simulation analysis based on a 3D grid map and a spherical robot moving with uniform velocity. The results show that compared with traditional path planning algorithms, the proposed algorithm can minimize the energy consumption and path length of the spherical robot as much as possible.

## 1. Introduction

As a class of robots, mobile robots play an important role in daily life and factory production. Different from wheeled or tracked robots, the spherical robot has a special driving mechanism. The spherical robot generates driving force by shifting its centroid, which controls the motion. The robot has a fully wrapped shell, protecting control modules, drive devices and sensors. The ball-shaped outer shell weakens the impact of environment significantly. The shell also serves as a part of the motion mechanism. Owing to the design, the spherical robot can survive in hostile environments such as swamps, grasslands and deserts and can also perform tasks in low spaces such as pipelines and underground power pipe galleries. In addition, the spherical robot has gained attention as a promising solution to address the challenges associated with planetary exploration. Considering the excellent adaptability of the spherical robot, this robot is considered to be one of the most suitable equipment for extraterrestrial exploration. Therefore, the spherical robot has broad application prospects. These working environments make autonomous operation necessary, and the path planning of spherical robots is the basis for achieving this goal. Different from indoor robots, spherical robots working outdoors must pay attention to energy consumption to increase working hours and avoid outages caused by power failure. However, existing research on the path planning of spherical robots only focuses on finding a passable path based on ideal conditions but ignores the problem of path energy consumption. Studying the energy consumption problem in path planning is the key to further expanding the application of spherical robots.

Path planning plays an important role in the motion and planning of robots. The autonomous motion of robots depends on good path planning results. Planned paths guide the movement of robots. Path planning can be divided into global planning and local planning. Also, most methods proposed in studies can be divided into search-based algorithms, sample-based algorithms and intelligent algorithms. Scholars have proposed lots of solutions, such as the Dijkstra algorithm [1], A* algorithm [2,3], JPS method [4], RRT algorithm [5], PSO algorithm [6], genetic algorithm [7], ant colony algorithm [8] and the artificial neural network [9]. These methods can realize the path planning under the environment with various obstacles. Moreover, most path planning algorithms are also suitable for solving the Traveling Salesman Problem (TSP), which is a simple path planning problem.

With the advancement of studies on the spherical robot, some works have discussed the path planning methods of spherical robots. In [10], Guo et al. proposed a spherical robot path planning method based on the fuzzy control algorithm. This method guides the spherical robot to reach the target position in an unknown environment successfully. In order to solve the deadlock problem in path planning, [11] proposed a step optimal path planning algorithm based on fuzzy control. By maintaining a certain distance from obstacles, this method improved the stability and sensitivity of spherical robots in an unknown environment. In the paper [12], a monocular vision system was installed on a spherical robot to explore the environment. This study proposed a path planning algorithm based on a single image. With the development of spherical robots, some scientists designed improved spherical robots, which can work underwater. For overcoming challenges in the water, Zheng et al. proposed a dynamic path planning algorithm based on a general constrained optimization problem (GCOP) model and a sequential quadratic programming (SQP) method with sensor input [13]. The algorithm improves the flexibility of spherical robots traveling in the water. Based on the characteristics of the spherical robot as a nonholonomic system, Joshi et al. created a simple path planning method based on the dynamics of spherical robots. In each step, only one Euler angle of the robot is changed. This method is computationally efficient [14]. After researching the dynamics of a spherical robot with a single pendulum, an improved hybrid A* path planning method was proposed in [15]. This hybrid A* algorithm reduces the expansion nodes based on the minimal rotation area, improves the efficiency and ensures the safety of the robot. Additionally, some studies based on evolutionary algorithms such as the genetic algorithm are proposed to solve the problem of path planning.

The above research proposed different methods of planning paths for the spherical robot. These works expand the application scenarios of spherical robots and lead the development towards intelligence. However, studies on the path planning problem of spherical robots are not sufficient. Most of the current research focusing on finding the shortest path ignores the impact of the terrain variation. For robots working indoors or with fixed routes, a large number of supply stations can be installed at fixed points for charging, and therefore, energy consumption is not an important issue. However, spherical robots are commonly required to work in harsh outdoor conditions, even on extraterrestrial planets. Under these conditions, supply stations are rare, resulting in difficulties when supplying spherical robots with energy. Due to complex terrain variation, path planning algorithms that only focus on path length may greatly increase the energy consumption of spherical robots. In the worst case, the ignorance of energy consumption can make spherical robots unable to reach the target. Therefore, the energy consumption of the spherical robot must be taken into consideration during path planning.

This paper proposes an improved A* algorithm for path planning of spherical robots. The optimization objective of this algorithm is to minimize both the energy consumption and path length of the spherical robot as much as possible. In the improved A* algorithm, an energy consumption estimation model (ECEM) is established based on the force analysis of spherical robots. Force analysis includes four conditions of moving on the ground and hill. Based on the force analysis, the model of motor torque can be established. Also, a distance estimation model (DEM) is established based on the improved Euclidean distance of the grid map.

## 2. Force Analysis of Spherical Robot Moving on Ground and Hill

### 2.1. Structure of a Spherical Robot

This paper conducts research based on a spherical robot called SBOT-001, which was designed by the authors’ team. This spherical robot consists of an energy system, a drive system, a control system, a navigation system, a communication system and a sensor system. Figure 1 gives two sectional views of SBOT-001 in which only half of the shell is visible to easily show the internal mechanism. As shown in Figure 1, SBOT-001 has a single pendulum and three motors in its shell. The weight of batteries in the energy system is large. In order to save space and ensure that the center of gravity is at the lower position inside the robot, the batteries are fixed in the pendulum frame as a part of the pendulum.

SBOT-001 has a similar driving mechanism to that which is widely used in single-pendulum spherical robots [16,17]. In SBOT-001, motors 1 and 2 are used to drive the pendulum forward and backward. Motor 3 is a steering motor, which drives the pendulum left or right. In order to drive the pendulum, motors 1 and 2 are equipped with gearboxes, which are used to increase the motor torque. The efficiency of the motor is related to the rolling speed of the robot and motor torque. This robot moves by changing the position of the pendulum. The gravity of the pendulum forms a torque that drives the spherical shell to rotate. In addition, SBOT-001 is equipped with a camera, a GNSS module and an IMU. SBOT-001 can complete missions, including outdoor exploration, target detection and environmental information collection by equipping different task loads.

As a mobile robot driven by an internal pendulum, the feature of the spherical robot moving on the ground is related to kinetics and its state. In order to analyze the motion and kinetics of the spherical robot, the model of spherical robot is simplified as a spherical shell connected with a pendulum. The top view and front view of the spherical robot motion are shown in Figure 2. The radius of the spherical shell is *r*, and the spherical center is point *Z*. In the motion, the angle between moving direction and *x*-axis is θ. The gravity of pendulum drives the spherical shell movement and controls the direction. Based on kinetics, detailed force analysis of the spherical robot can be carried out.

### 2.2. Force Analysis of Spherical Robot

In the following analysis, some unimportant quantities such as framework weight, electronics weight and air resistance are ignored. Mechanical losses between components of the spherical robot are not considered. In this paper, all angles are measured in a counterclockwise direction. Assuming that the robot is only affected by the external forces of gravity and friction, the kinetics of the spherical robot can be analyzed by using the laws of physics. The movement of the spherical robot is continuous in the whole movement process. There are only a few cases where the spherical robot needs to change its moving direction. Frequent changes in speed lead to higher energy consumption. So, motion can be seen as a constant process, and the spherical robot keeps a uniform velocity. In addition, the friction coefficient between the spherical robot and the ground is also a constant.

When the spherical robot moves on the ground and hill, Figure 3 shows the force analysis of the robot. This paper gives an analysis of four conditions according to the terrains and the friction. In all conditions, the spherical robot moves at a uniform speed, and the rolling resistance is an obstacle to the movement. In Figure 3, picture (a) gives the force analysis of moving on the flat ground. Under this condition, the spherical robot needs a power-assist to overcome the rolling resistance. Picture (b) gives the force analysis of moving uphill where the robot also needs a power-assist to overcome the resistance from gravity and rolling resistance. However, in pictures (c) and (d), gravity is the active force, which drives the robot to roll down. In picture (c), the friction is large, and the pendulum needs to turn forward because the gravity component of shell cannot overcome the rolling resistance. In picture (d), the friction is small, and the pendulum needs to turn backward because the gravity component of shell is bigger than the rolling resistance. In order to keep the robot rolling at a uniform speed, the motors must output torque to prevent acceleration and keep the velocity in pictures (a) and (b).

In order to make the description of the analysis clear, several parameters have been used in this section. *M* and *m* are the weight of the spherical shell and the weight of the pendulum. *r* and *l* represent the radius of the spherical shell and the length of the pendulum. Parameter *g* is the gravity coefficient. $T_{M}$ is the torque output by motors. Force $F_{L}$ is the tension between the pendulum and shell. Point *P* is the contact point between the spherical shell and the ground. Parameter *α* is the slope angle of the ground. In force analysis, *α* is also the angle between the gravity direction and the line that connects point *P* and the sphere center. In motion, the pendulum swings at an angle called *β*, which is the angle between the pendulum and the gravity direction. Parameter $l_{1}$ is the distance between the plumb line at the sphere center and the plumb line at point *P*. Parameter $l_{2}$ is the distance between the plumb line at the sphere center and the plumb line at the center of the pendulum. Parameter *δ* is the rolling resistance coefficient, which is used to calculate rolling resistance.

In this section, the velocity of the robot is a constant. Two force balance equations can be established by using Newton’s second law. These equations are expressed as:(1)$$\begin{array}{l} \frac{T_{M}}{l}\sin(\beta )+F_{L}\cos(\beta )=mg\\ \frac{T_{M}}{l}\cos(\beta )=F_{L}\sin(\beta ) \end{array}$$
where $T_{M}/l$ is the force, which is produced by the main motors and acts on the pendulum. The direction of $T_{M}/l$ is perpendicular to the pendulum, as shown in Figure 3. By solving the above equations, the following results can be obtained.
(2)$$\begin{array}{l} F_{L}=mg\cos(\beta )\\ T_{M}=mgl\sin(\beta ) \end{array}$$

Further, the pressure of the robot on the ground, called $F_{N}$, can be obtained in different conditions. $F_{N}$ includes $F_{N1}$, $F_{N2}$, $F_{N3}$ and $F_{N4}$.
(3)$$\begin{array}{l} F_{N1}=Mg+mg\cos(\beta )\cos(\beta )\\ F_{N2}=Mg\cos(\beta )+mg\cos(\beta )\cos(\beta -\alpha )\\ F_{N3}=Mg\cos(\alpha )+mg\cos(\beta )\cos(\beta +\alpha )\\ F_{N4}=Mg\cos(\alpha )+mg\cos(\beta )\cos(\beta -\alpha ) \end{array}$$
where $F_{N1}$ is the pressure when the robot moves on the flat ground, $F_{N2}$ is the pressure when the robot moves uphill and $F_{N3}$ and $F_{N4}$ are the pressures when the robot moves downhill. $F_{N3}$ is the pressure when the friction is large and the pendulum needs to turn forward. In contrast, $F_{N4}$ is the pressure when the friction is small and the pendulum needs to turn backward. In the analysis, *α* and *β* are measured in a counterclockwise direction. In Figure 3, directions of all angles are represented. Specifically, *α* is zero, and *β* is a positive value in picture (a). In picture (b), *α* and *β* are both positive values. In picture (c), *α* is a negative value and *β* is a positive value. In picture (d), *α* and *β* are both positive values. 

Based on the properties of trigonometric function, equations in Equation (3) can be unified as:(4)$$F_{N}=Mg\cos(\alpha )+mg\cos(\beta )\cos(\beta -\alpha )$$

Furthermore, based on the torque balance theorem, torque balance equations of four conditions can be established as:(5)$$\begin{array}{l} \delta F_{N1}=mgl\sin(\beta )\\ \delta F_{N2}+Mgr\sin(\alpha )=mg\left(l\sin(\beta )-r\sin(\alpha )\right)\\ \delta F_{N3}=mg\left(l\sin(\beta )+r\sin(\alpha )\right)+Mgr\sin(\alpha )\\ \delta F_{N4}+mg\left(l\sin(\beta )-r\sin(\alpha )\right)=Mgr\sin(\alpha ) \end{array}$$

In the above equations, $\delta F_{N1}$, $\delta F_{N2}$, $\delta F_{N3}$ and $\delta F_{N4}$ are the rolling resistance whose direction is always against moving direction. Considering that *α* and *β* are signed numbers, the equations in Equation (5) can also be unified. The unified equation of torque balance can be proposed as:(6)$$\delta F_{N}+Mgr\sin(\alpha )=mg\left(l\sin(\beta )-r\sin(\alpha )\right)$$

In a 3D map, *α* can be acquired from the map date. The value of *β* can be obtained by solving the set consisting of Equations (2), (4) and (6). In the process of solving, cosine function cos(β) should be converted to sine function in order to simplify the process. The motor torques can be estimated when the value of *β* is clear. Bringing *β* into Equation (2), we can get the value of torque $T_{M}$ as following:(7)$$T_{M}=(\delta +r)Mg\sin(\alpha )+mgr\sin(\alpha )+\delta F_{L}\cos(\beta -\alpha )$$

In addition, there are some points that the spherical robot cannot reach. These points have huge height differences with neighbor points and exceed the upper limit of dynamics. No matter how large the motor output is, the largest torque that drives the spherical shell rolling is produced when the value of *β* is π/2 or −π/2. The maximum driving force cannot overcome the resistances.

## 3. Improved A* Algorithm of Spherical Robot Considering Energy Consumption

### 3.1. Traditional A* Algorithm

The A* algorithm is a path planning algorithm widely used for robots. The A* algorithm can be seen as an extension of the Dijkstra algorithm. It creates an open list and a closed list to store child points and parent points based on the link list. This algorithm makes the planned result purposeful by establishing a heuristic function between the start point and the target point. The heuristic function f(n) can be calculated as follow:(8)f(n)=G(n)+H(n)
where parameter *n* is the number of the current point in grid map. The G(n) is the actual distance from the start point to the current point based on the closed list. The H(n) is the function that estimates the distance from the current point to the target point based on Euclidean distance. Assume that the start point is point 1, whose coordinate is $\left(x_{1},y_{1}\right)$, the target point is point *z*, whose coordinate is $\left(x_{z},y_{z}\right)$ and the current point is point *n*, whose coordinate is $\left(x_{n},y_{n}\right)$. In addition, the points that make up the path are denoted by point 1 to point *n*. G(n) and H(n) can be shown as:(9)$$\begin{array}{l} G(n)=\sum_{i=1}^{n-1}d_{i,i+1}\\ H(n)=\left|\sqrt{{\left(x_{n}-x_{z}\right)}^{2}+{\left(y_{n}-y_{z}\right)}^{2}}\right| \end{array}$$
where $d_{i,i-1}$ is the distance between point *i* and point (*i* + 1). Additionally, G(n) decides the selection of points, and H(n) guides the path to find the target. In the process of finding the path, the A* algorithm selects the grid with the lowest cost value as the extended grid in each step. The traditional A* algorithm focuses on finding the shortest path between the start point and the target point. This algorithm finds the shortest path through continuous iteration and traversal until finding the path.

### 3.2. Slope Angle Calculation on a 3D Map

Different from the 2D map, the variation of terrain on the 3D map makes path planning difficult. In order to use the A* algorithm for path planning on a 3D map, the slope angle *α* between the points on the 3D map needs to be calculated.

The 3D map is simplified to a grid map, which adds another dimension compared to a 2D grid map. The extra dimension makes the 3D grid map have more information [18,19]. Additionally, 3D maps offer height data that can be used to calculate height differences, distances and angles. For example, the height data of point *n* is shown as h(n). Different from traditional 2D grid maps, which are widely used in path planning methods, 3D grid maps rarely appear in path planning, especially in studies of ground mobile robots. As a kind of ground mobile robot, the spherical robot can only change the *z*-axis position with the variation of terrain. Based on the fact, the 3D grid maps used in this paper can be seen as 2D grid maps with height information such as that shown in Figure 4, which means these 3D maps have the same characteristics as 2D grid maps. When the spherical robot moves on the grid map, there are eight directions to be selected. Compared to 3D maps used in path planning methods for drones or AUVs [20,21,22], the 3D maps used in this paper reduce the complexity and computing burden of planning.

The height difference is a signed number. Based on the grid map shown in Figure 4a, the height difference from point *n* $\left(x_{n},y_{n}\right)$ to point (*n* + 1) $\left(x_{n+1},y_{n+1}\right)$ and the distance between these two points can be calculated as:(10)$$\begin{array}{l} h_{n,n+1}=h(n)-h(n+1)\\ d_{n,n+1}=\sqrt{h_{n,n+1}^{2}+{\left(x_{n+1}-x_{n}\right)}^{2}+{\left(y_{n+1}-y_{n}\right)}^{2}} \end{array}$$
where $h_{n,n+1}$ is a signed number presenting the height difference from point A to point B, and $d_{n,n+1}$ is an unsigned number presenting the Euclidean distance between these two points. The slope angle from point *n* to point (*n* + 1) can be calculated by the inverse trigonometric function:(11)$$\alpha_{n,n+1}=\arctan(h_{n,n+1}/d_{n,n+1})$$
where $\alpha_{n,n+1}$ is a signed number and ranges from −π to π. Additionally, there are eight potential directions for the robot to select, as shown in Figure 4b. The selected direction connects the current point with the next neighbor point. 

### 3.3. Energy Consumption Estimation Model

In order to count the energy consumed by the spherical robot on the path, an energy consumption estimation model (ECEM) is established based on the slope angle and force analysis. ECEM ignores the loss in transmission process and tiny resistances. The energy consumption of turning at each point is very small and has a small influence on planning, so it is overlooked in the ECEM. In addition, the friction coefficient between the spherical robot and the ground is a constant.

Assume that point *n* $\left(x_{n},y_{n}\right)$ and point (*n* + 1) $\left(x_{n+1},y_{n+1}\right)$ are neighbor points on the grid map. According to the above section, the angle between two points can be expressed as $\alpha_{n,n+1}$. Based on the force analysis mentioned in Section 2.2, the motor torque can be shown as $T_{M}(\alpha_{n,n+1})$, which is related to the angle $\alpha_{n,n+1}$. The motor power between point *n* and point (*n* + 1) can be calculated as follows:(12)$$P_{n,n+1}=Fv=\left(T_{M}(\alpha_{n,n+1})/r\right)v/\eta (T_{M},v)$$
where $\eta (T_{M},v)$ is the motor efficiency, which is affected by the motor speed *v* and motor torque $T_{M}(\alpha_{n,n+1})$. The planned path consists of the parent nodes in the closed list. These parent nodes correspond to the points on grid map. Based on the energy calculation formula, the energy consumption between two adjacent points is calculated as:(13)$$e_{n,n+1}=P_{n,n+1}t=P_{n,n+1}\left(d_{n,n+1}/v\right)$$
where *t* is the time required for the robot to move from point *n* to point (*n* + 1). The total energy consumption can be estimated by the sum of the energy consumption between two adjacent points on the path. If the number of points in the path is *n* and they are recorded as point 1 to point *n*, the equation of ECEM can be proposed as:(14)$$E(n)=\sum_{i=1}^{n-1}e_{i,i+1}$$

### 3.4. Distance Estimation Model

In this section, a distance estimation model (DEM) is established based on the 3D grid map to estimate the distance between two points. Equation (10) has given the distance between two points based on Euclidean distance algorithm. However, the moving direction of the spherical robot can only be selected from the specified eight directions, which are mentioned in Section 3.2. Traditional Euclidean distance ignores the direction limitation. If the target point is far from the current as shown in Figure 5, there could be a fantastic error between the Euclidean distance and the real value. The error may cause the planning result to be far from expectations.

For estimating the distance between two points accurately, the DEM is proposed. The DEM is an improved Euclidean distance model to estimate distance in a grid map. Assume that the blue point in the Figure 5 is the current point *n* whose coordinate is $\left(x_{n},y_{n}\right)$ and the red point is the target point *z* whose coordinate is $\left(x_{z},y_{z}\right)$. The DEM is proposed as:(15)$$H^{\prime }(n)=\sqrt{2}\min\left(\left|x_{z}-x_{n}\right|,\left|y_{z}-y_{n}\right|\right)+\left|\left|x_{z}-x_{n}\right|-\left|y_{z}-y_{n}\right|\right|$$

Compared with traditional Euclidean distance, DEM obeys the requirement of moving direction and is more matched with the planned path in a grid map.

### 3.5. Improved A* Algorithm Considering Energy Consumption

The A* algorithm is widely used in path planning. However, the traditional A* algorithm only focuses on the length of path and ignores the energy consumption. It is not easy for the spherical robot to find a supply station for charging power outdoors. Energy consumption is an important research topic regarding spherical robots. Path planning with the goal of minimum energy consumption is an effective method to reduce energy consumption. In order to find the path with low energy needs and save energy, this paper proposes an improved A* algorithm for the path planning of the spherical robot, considering energy consumption.

The proposed algorithm has a new heuristic function containing ECEM and DEM, which are mentioned above. ECEM estimates the energy consumption between the start point and the current point. DEM gives a reference about the distance between the current point and the target point. These two models guide the algorithm to find an optimal path connecting the start point with the target point. Compared with the traditional A* algorithm, the proposed new algorithm takes both distance and energy consumption into consideration.

The flow chart of the improved A* algorithm and the traditional A* algorithm is illustrated in Figure 6. The pseudocode of the proposed algorithm is illustrated in Algorithm 1.

**Algorithm 1** Improved A* Algorithm for Path Planning of Spherical Robot Considering Energy Consumption
**/*Initialization*/**
Load data of the 3D grid map and parameters of the spherical robot;Set the start point $\left(x_{1},y_{1}\right)$ and the target point $\left(x_{z},y_{z}\right)$;Create an open list and a closed list;Introduce the start point into the open list and initialize the closed list empty;The number of points in the open list is *a*, and the number of points in the closed list is *b*;

**/*Iterative search*/**
6.**while** *a* ≠ 07.Find the next point *n* $\left(x_{n},y_{n}\right)$ with the minimum *f*(*n*) in the open list, remove it from open list and add it into the closed list;8.**if** ($x_{n}\neq x_{z}$) or ($y_{n}\neq y_{z}$)9.Add all passible neighbor points of point *n* into a set called subs.10.**if** subs is empty11.Go to **Step 6**;12.
**end if**
13.Select a new point (*n* + 1) from subs.14.Calculate distance and angle between point n and point (*n* + 1).15.Calculate motor torque $T_{M}$ based on force analysis.16.Calculate the energy consumption of temporary path based on ECEM.17.**if** point (*n* + 1) is in the open list18.**if** there is lower cost to reach point (*n* + 1)19.renew *f*(*n*);20.
**end if**
21.
**end if**
22.**if** point (*n* + 1) is in the open list23.**if** there is lower cost to reach point (*n* + 1)24.renew *f*(*n*);25.
**end if**
26.
**end if**
27.Add point (*n* + 1) into open list;28.Calculate $H^{\prime }(n+1)$ based on DEM;29.Calculate $f(n+1)=kE(n+1)+H^{\prime }(n+1)$;30.Go to **Step 6**;31.
**end if**
32.
**end while**
33.Add point *n* into closed list;34.Find parent points in closed list one by one;35.Output the best path and its cost value;36.
**Post-process results and visualization**



There are four differences between the two algorithms. Firstly, the proposed algorithm loads the data of a 3D grid map, while the traditional algorithm loads 2D map information. Secondly, the cost function that decides the next point is different in the two algorithms. In the traditional A*, the next point (*n* + 1) is decided by the cost function G(n+1), which represents the template path length from the start point to the point (*n* + 1). But the cost function in the improved A* has been changed to the E(n+1), which represents the energy consumption from start point to the point (*n* + 1). Thirdly, the proposed algorithm establishes the $H^{\prime }(n+1)$ based on DEM, while the traditional A* establishes the H(n+1) according to the Euclidean distance. Finally, the traditional A* algorithm builds a heuristic function only based on distance information. The improved A* algorithm takes both ECEM and DEM into the new heuristic function. Assuming the current point is *n*, the new heuristic function can be shown as:(16)$$f(n)=kE(n)+H^{\prime }(n)$$
where E(n) is the energy consumption estimation model (ECEM), and $H^{\prime }(n)$ is the distance estimation model (DEM). Parameter *k* is the weight parameter, which changes the importance of ECEM in the heuristic function. The heuristic function plays a key role in the proposed algorithm. In the initial stage of path planning, the value of *E*(*n*) is tiny, while *H*′(*n*) plays a major role and guides the algorithm to search for the target point. With the advancement of the planning, the current point gradually approaches the target. The estimated energy consumption increases, the estimated distance decreases, and the influence of ECEM increases. This phenomenon makes the choosing of path points depend not only on the distance estimate but also on the energy consumption. As the distance between the current point and the target point decreases, *E*(*n*) will occupy more weight. In addition, *k* also affects the performance of the algorithm. When the value of *k* is large, the improved A* algorithm degenerates to the Dijkstra algorithm that aims to find the path with smallest energy consumption and ignores the influence of path length. On the contrary, it is a greedy algorithm that only wants to find the shortest path connecting two points.

## 4. Simulations and Discussion

In order to test the feasibility and effectiveness of the improved A* algorithm in this paper, a virtual experiment environment is established. The experimental simulation platform was a Windows 10 computer with Intel i5-12600KF CPU, 16 GB RAM and software MATLAB 2022b.

### 4.1. Simulation Scene

In this paper, the force analysis is based on the spherical robot moving at a uniform speed. Suppose that the spherical robot keeps a uniform velocity whether it moves on the ground or a hill. The velocity stays at 1 m/s. The whole map is passible except for points that exceed the limit of dynamics. In order to keep the status, the motors stay active all the time. The spherical robot needs motor torque to overcome obstacles, including the rolling resistance. The rolling resistance coefficient *δ* is selected as 0.05 in this research. The gravity coefficient *g* is 9.8 m/s^2^ in this simulation. The total consumption can be estimated by summing energy consumptions between path points. 

The spherical robot mentioned in Section 2.1 has several important parameters: the weight of spherical shell *M* is 1 kg, the weight of pendulum *m* is 3 kg, the radius of the spherical robot *r* is 20 cm and the length of pendulum *l* is 15 cm. The spherical robot proposed in this paper has motor 1 and motor 2 to control its speed. The variation of motor efficiency *η* is shown in Figure 7. Figure 7 is a motor efficiency map that reveals the relationship between motor efficiency and torque, as well as the rolling speed of the robot. This map chart can be obtained through making tests on a motor called MGL42GP-775. It should be noted that the motor torque proposed in this paper is the sum of the two motor torques, and the variation of the *η* is only affected by motor torque when the speed of the spherical robot is a constant. In this paper, the velocity is equal to the rolling speed. The value of the rolling speed is 1 m/s, and the *η* is only affected by the torque. In Figure 7, the variation of motor efficiency at a speed of 1 m/s has been marked with a red line. In the following content, the energy consumption estimation will be based on this condition.

In order to verify the effectiveness of the improved A* algorithm proposed in this paper, the terrains in the surrounding environment are converted to a grid map as shown in Figure 8. The grid map can be described as a matrix whose elements are height data. The variation of color presents the changing of height. In this map, each grid is a cube with a side length of 1 m. In order to make the environment more complex, a rectangular area that is out of the dynamics limits has been added into the 3D map. The height of the rectangular area is discontinuous compared with the surrounding area.

### 4.2. Simulation Using Traditional A* Algorithm

In order to verify the validity of A* algorithm in a 3D grid map, set the start point to (3, 28) and the target point to (29, 9). Then, apply the traditional A* algorithm, and find a valuable path in the map. The path planned by the traditional A* algorithm is shown in Figure 9. In Section 3.2, this paper gives the description of a 2D grid map containing height data. The 2D grid map is another expression form of the 3D map. In order to show the process of path planning, some results are shown in Figure 9b. In picture (b), the 2D grid map has 30 columns and 30 rows. Each grid is a square with a side length of 1 m. In this map, white grids are the passible points, light gray grids are child points in open list to be selected, dark gray grids are parent points in closed list which has been searched and the black grids composing a passable path. The path connected point (3, 28) with point (29, 9). The result of the planning shows that the planned path length is about 35.314 m, the total consumption is about 614.196 joules and the closed list has 174 points.

Motors output torques during the movement. Figure 10 shows the variation of the torque. The var of torque during motion is about 2.1167.

### 4.3. Simulations Using Improved A* Algorithm

In order to prove the validity and advantages of the proposed path planning algorithm, set the start point as (3, 28) and the target point as (29, 9). Then, apply the improved A* algorithm, and find a valuable path in the map. In this simulation, the *k* is set as 1. Figure 11 shows the result based on the proposed algorithm. The result shows that the length of planned path is about 36.731 m, the total consumption is about 552.432 joules and the closed list has 837 points. The reduction of energy consumption using the improved algorithm is about 10.1%, which is a great advantage compared with traditional A*. 

Motors output torques during the movement. Figure 12 shows the variation of the torque. The var of torque during motion is about 1.1987. The smaller variance of torque indicates that the terrain of path planned by improved A* is flatter, which is also shown in the 3D grid map.

The reduction of the energy consumption using the proposed algorithm is about 10.1% compared with the traditional A* algorithm. The detailed comparison of the two experimental results is shown in Table 1. Table 1 demonstrates that the improved A* algorithm brings a substantial reduction in energy consumption at the cost of a tiny increase in path length. The proposed algorithm in this paper is an effective method to reduce energy consumption of the planned path.

For exploring the law of bi-objective optimization, this paper changes the value of *k* from 0.1 to 0.3 and analyzes the variation of several important indexes. Figure 13 shows the variation and reveals that the path length and energy consumption decrease with the increase of *k* generally. The variation is small when the value of *k* is close to 0.3. This phenomenon indicates that the optimal result of the path planning using improved A* algorithm can be achieved when the value of *k* is bigger than 0.3. Of course, all the results given by the proposed algorithm have an advantage over the results given by the traditional one. Lower energy consumption makes the spherical robot have more hours to work outdoors without an energy supply.

All simulations are conducted in the environment that is mentioned at the beginning of this section. The time cost of each simulation presents the convergence speed. The time costs of simulations with different *k* are recorded in this paper. Details are shown in Table 2. With the increase of *k*, the time cost and number of points in the closed list increase at the same time. This phenomenon indicates that the increase in *k* enhances the weight of ECEM in the proposed improved A* algorithm. This enables the proposed algorithm to place greater emphasis on energy consumption. However, the increase in *k* also brings many more points into the potential choices. It makes the proposed algorithm spend more time dealing with the calculations and comparisons. With the increase in *k*, the efficiency of the improved A* algorithm weakens.

As shown in Figure 14, the effectiveness and efficiency of the proposed algorithm are opposed. It should be mentioned that all the results are influenced by the map data, and the variation of the terrains plays the key role in the planning. In fact, the value of *k* needs to be adjusted according to the situation and requirement when the proposed algorithm is deployed on a real spherical robot.

## 5. Conclusions and Future Work

This paper proposes an improved A* algorithm for the path planning of a spherical robot, considering energy consumption. In the proposed algorithm, an energy consumption estimation model (ECEM) is established based on the force analysis, and a distance estimation model (DEM) is established based on the improved Euclidean distance. The effectivity of the proposed algorithm has been proved through several simulations. Compared with the traditional A* algorithm focusing on path length, the proposed algorithm minimizes both energy consumption and path length as much as possible. As the weight of ECEM increases, the convergence speed of improved A* algorithm slows down. The proposed algorithm takes energy consumption into the path planning of spherical robots firstly and gives a method to estimate energy consumption in 3D maps. In practical applications, using this algorithm reduces the risk of shutdown due to insufficient energy and allows more energy to be used for tasks rather than travel. This work gives guidance for the following applications of the spherical robot.

Although the improved A* algorithm proposed in this paper has advantage in saving energy compared with the traditional A*, there is still room for improvement. The ECEM used in this paper is established in an ideal state without considering the influence of variable speed and transient response. In addition, the proposed algorithm needs to know the whole map in advance and is unsuitable for dynamic planning during moving. In future works, path planning focusing on the dynamic surrounding environment will be researched. Studies regarding decision-making and planning based on external environment and internal states will be performed for the spherical robot. 

## Figures and Tables

**Figure 1 sensors-23-07115-f001:**
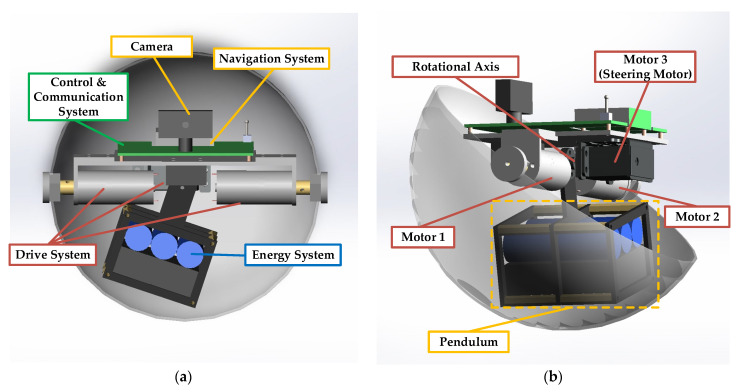
Sectional views of SBOT-001: (**a**) Front view of the spherical robot and description; (**b**) Side view of the spherical robot and description.

**Figure 2 sensors-23-07115-f002:**
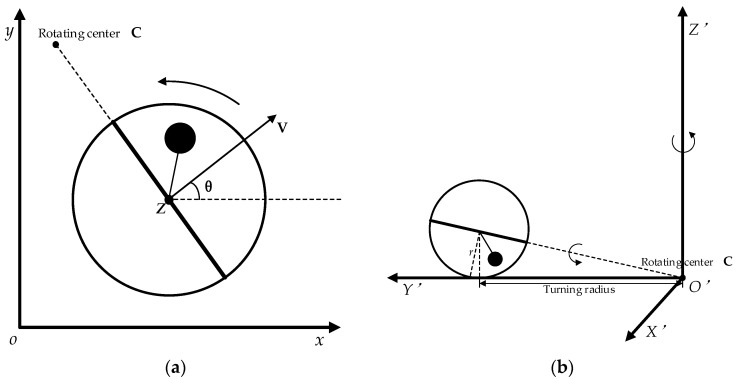
Top and front view of spherical robot moving turn left: (**a**) Top view of spherical robot and motion description; (**b**) Front view of spherical robot and motion description.

**Figure 3 sensors-23-07115-f003:**
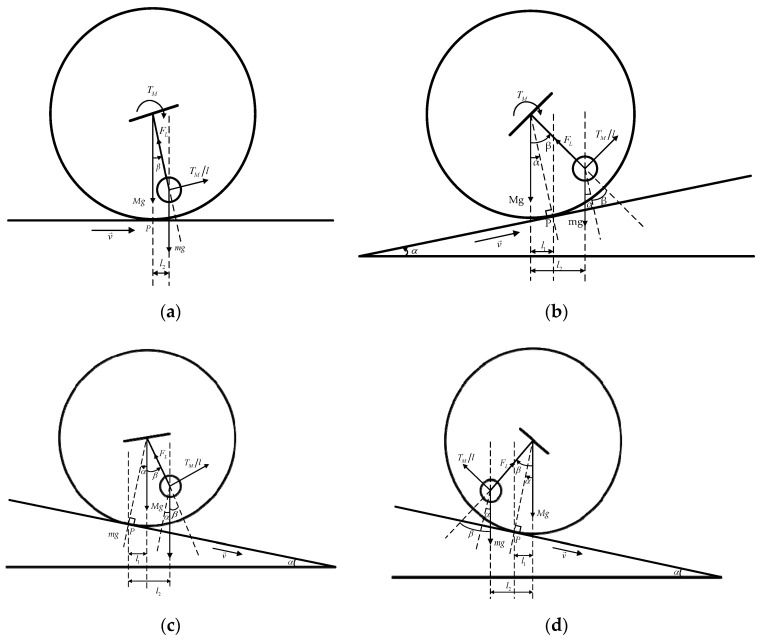
Spherical robot moving on the ground and hill: (**a**) Analysis of spherical robot moving on a flat ground. (**b**) Analysis of spherical robot moving uphill. (**c**) Analysis of spherical robot moving downhill when the friction is large. (**d**) Analysis of spherical robot moving downhill when the friction is small.

**Figure 4 sensors-23-07115-f004:**
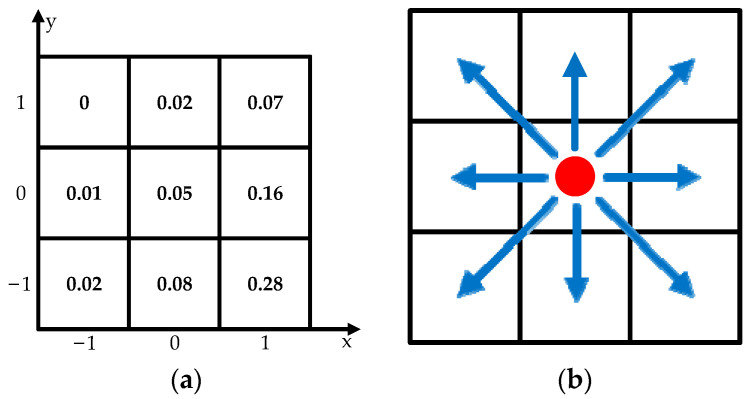
2D grid maps with height information and directions for choosing: (**a**) A 2D grid map which contains height data in each grid; (**b**) Potential moving directions when spherical robot stays at the red point.

**Figure 5 sensors-23-07115-f005:**
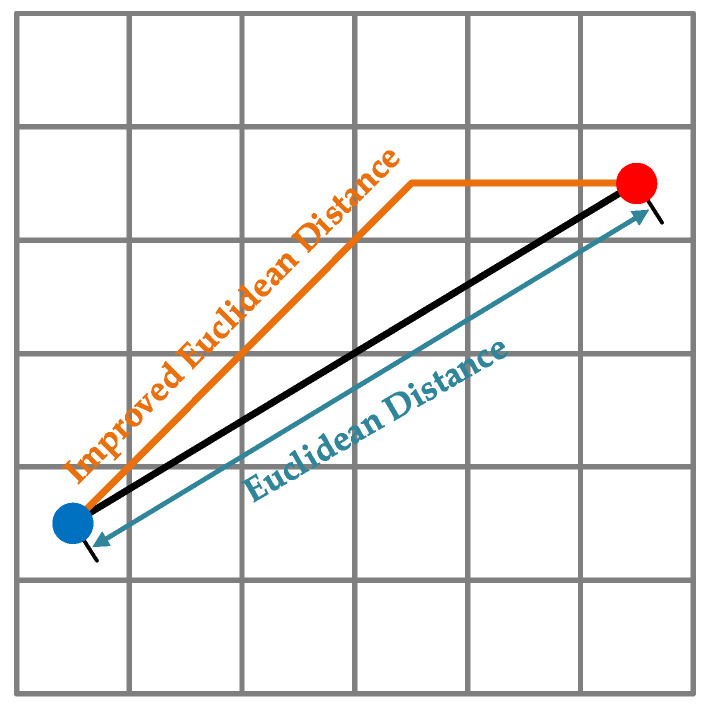
The Euclidean distance and improved Euclidean distance based on grid map.

**Figure 6 sensors-23-07115-f006:**
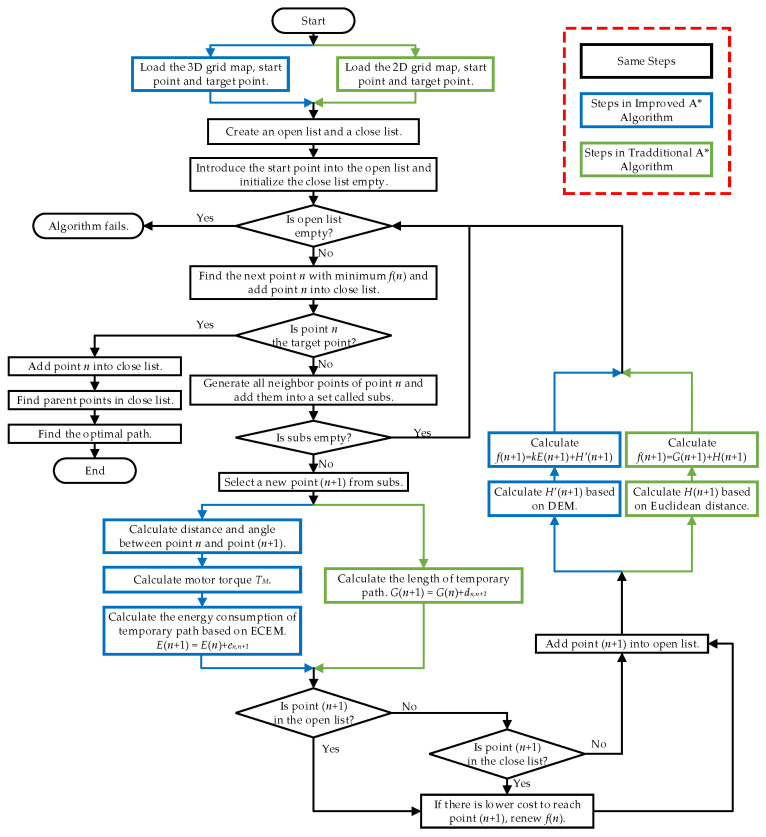
The flow chart of the improved A* algorithm and the traditional A* algorithm.

**Figure 7 sensors-23-07115-f007:**
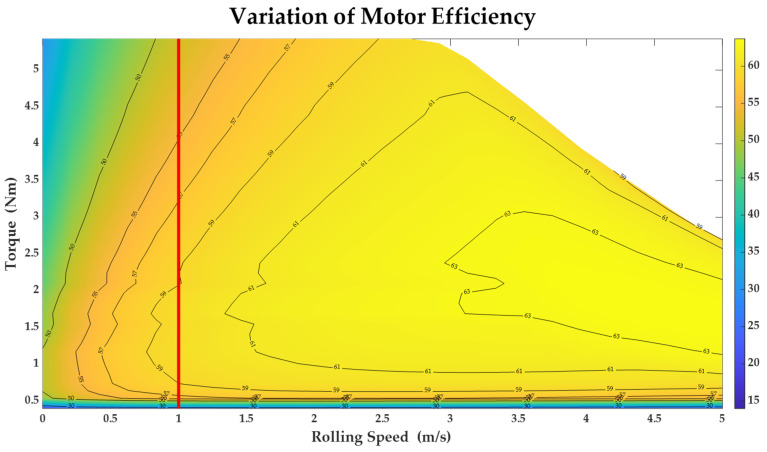
Variation of motor efficiency related to motor torque and rolling speed.

**Figure 8 sensors-23-07115-f008:**
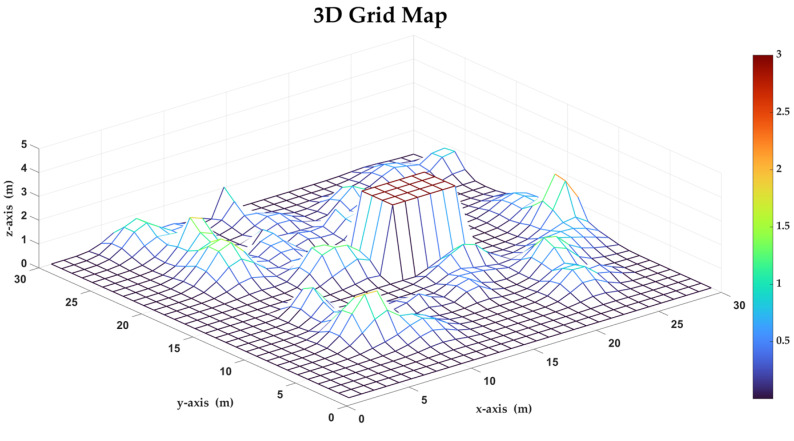
The 3D gird map used in this paper.

**Figure 9 sensors-23-07115-f009:**
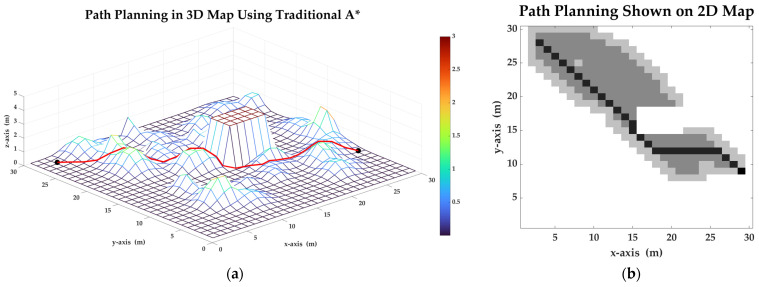
Path planning in a 3D map using traditional A* algorithm: (**a**) The planned path on the 3D map using traditional A* algorithm. (**b**) The planned path focusing on distance shown on the 2D grid map.

**Figure 10 sensors-23-07115-f010:**
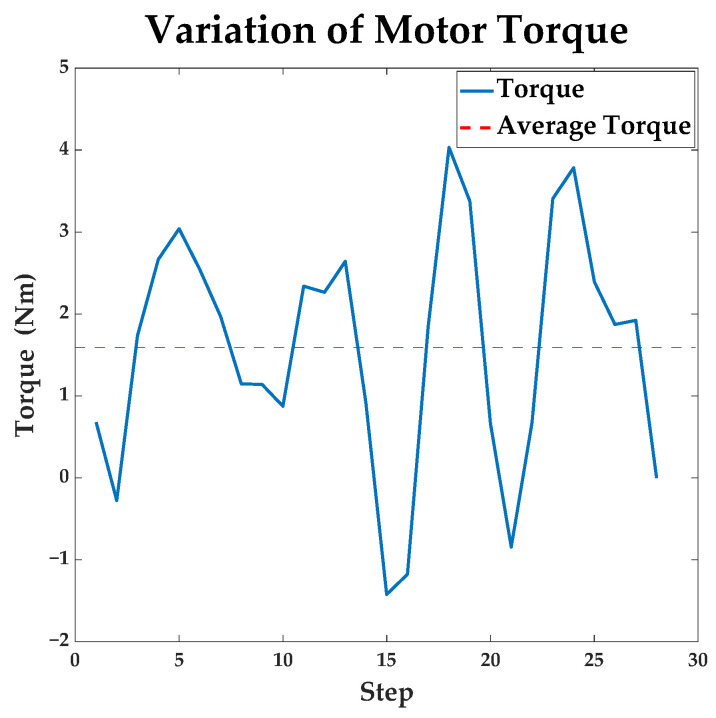
The variation of motor torque in the path planned by the traditional A* algorithm.

**Figure 11 sensors-23-07115-f011:**
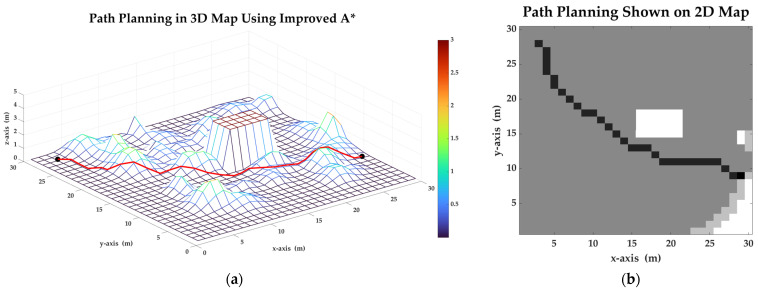
Path planning in a 3D map using improved A* algorithm with *k* = 1: (**a**) The planned path considering energy consumption on the 3D map. (**b**) The planned path considering energy consumption shown on the 2D grid map.

**Figure 12 sensors-23-07115-f012:**
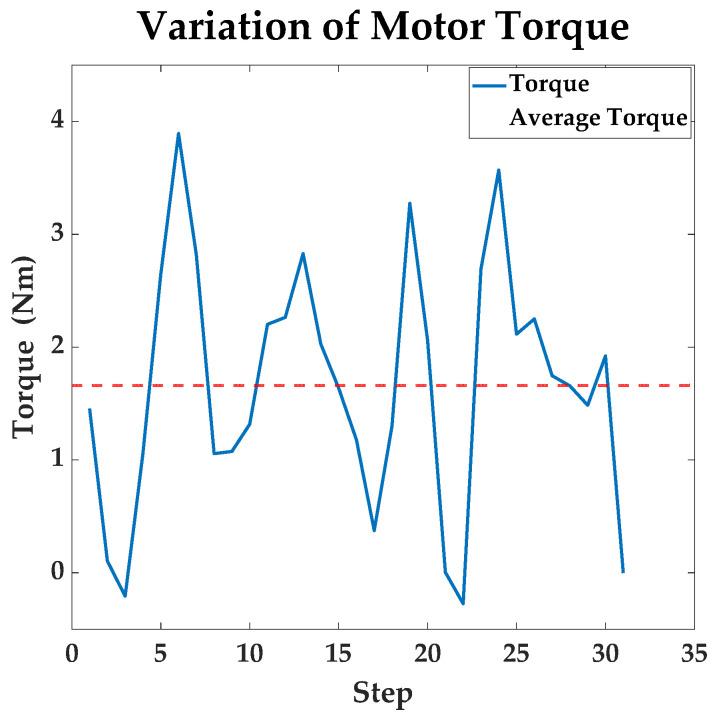
The variation of motor torque in the path planned by the improved A* algorithm.

**Figure 13 sensors-23-07115-f013:**
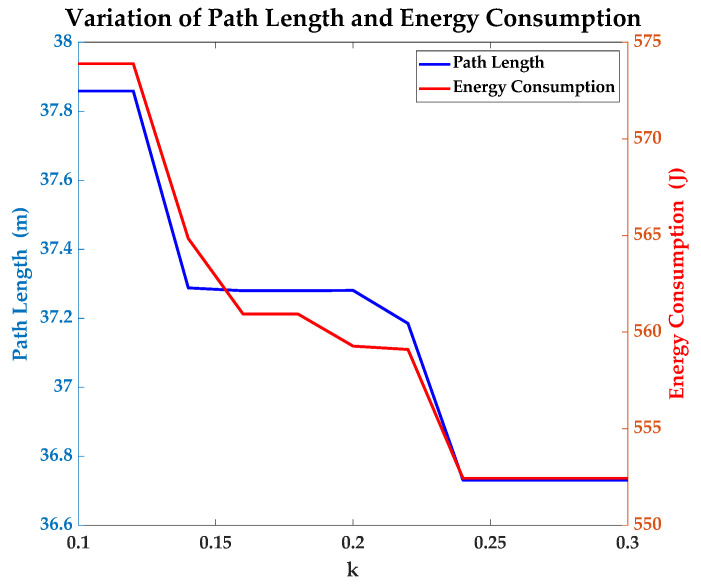
Variation of path length and energy consumption under different values of *k*.

**Figure 14 sensors-23-07115-f014:**
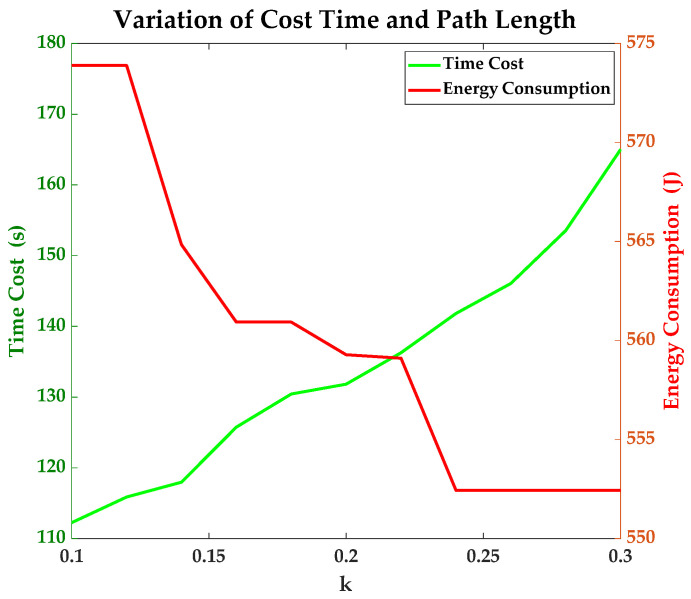
Variation of cost time and energy consumption under different values of *k*.

**Table 1 sensors-23-07115-t001:** The comparison of using traditional A* and using improved A* with *k* = 1.

	Traditional A* Algorithm	Improved A* Algorithm	Comparison
Path length	35.314 m	36.731 m	+4.0%
Energy consumption	614.196 J	552.432 J	−10.1%

**Table 2 sensors-23-07115-t002:** The variation of several results under different values of *k*.

*k*	Path Length (m)	Energy Consumption (J)	Closed list Points Number	Time Cost (s)
0.10	37.859	573.891	600	112.229169
0.12	37.859	573.891	667	115.875186
0.14	37.288	564.846	703	117.951924
0.16	37.280	560.933	739	125.747523
0.18	37.280	560.933	764	130.431161
0.20	37.281	559.278	773	131.826470
0.22	37.185	559.104	796	136.268961
0.24	36.731	552.432	794	141.816453
0.26	36.731	552.432	799	146.055316
0.28	36.731	552.432	802	153.551163
0.30	36.731	552.432	811	165.043557

## Data Availability

Not applicable.

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
