# Peer review of "Improved A* Algorithm for Path Planning of Spherical Robot Considering Energy Consumption"

_sensors, 2023, doi:10.3390/s23167115_

Round 1

Reviewer 1 Report

The A* Search is an informed best-first search algorithm that efficiently determines the lowest cost path between any two nodes in a directed weighted graph with non-negative edge weights. This paper proposes an improved A* algorithm for path planning of spherical robot. The optimization objective of this algorithm is to minimize both the energy consumption and path length of the spherical robot as much as possible. In the improved A* algorithm, an energy consumption estimation model (ECEM) is establish based on the force analysis of spherical robot. Force analysis includes four conditions of moving on the ground and hill. Based on the force analysis, the model of motor torque can be established. Also, a distance estimation model (DEM) is established based on the improved Euclidean distance of the grid map. The ideas of this paper have good reference values for this area researchers.

However, according to the paper's description 'in order to test the feasibility and effectiveness of the improved A* algorithm, a virtual experiment environment is established. Experimental simulation platform: Windows 10 computer with Intel i5-12600KF CPU, 16GB RAM and software MATLAB 2022b', and then draw the conclusion that 'The proposed algorithm in this paper is an effective method to reduce energy consumption of the planned path'. Maybe the conclusion has no doubt and however the real cases may be much more challenge, and the paper should consider more constraints of the true situation.

Addition, the paragraph line 95 to 104 seems unnecessary for whole paper, as  feels it is generated by AI technology and still require careful rearranged.

So, I suggest that this paper better be reorganized and resubmited.

Reviewer 2 Report

Overall, this manuscript covers an interesting topic on path planning for spherical robots, focusing on energy consumption optimization. Here are some suggestions to improve the manuscript:

Introduction: Expand the introduction to provide more context and motivation for the study. Explain why path planning for spherical robots is important and the significance of considering energy consumption in the process. Provide a brief overview of existing path-planning algorithms for spherical robots and highlight the limitations they face.

Literature Review: Include a section that summarizes the current state of the art in path planning for spherical robots. Discuss the strengths and weaknesses of existing algorithms, particularly those that focus on energy consumption and path length optimization.

Methodology: Provide a detailed explanation of the proposed improved A* algorithm considering energy consumption. Describe the key components of the algorithm and how the energy consumption estimation model (ECEM) and distance estimation model (DEM) are constructed. Include equations or pseudocode, if applicable, to help readers understand the algorithm better.

Heuristic Function: Clarify how the heuristic function combining ECEM and DEM works to determine the path cost for the A* algorithm. Explain the rationale behind choosing these particular models and their relevance to the problem.

Conclusion: Summarize the main contributions and findings of the study in the conclusion section. Emphasize the significance of the proposed algorithm for practical applications of spherical robots in real-world scenarios. Limitations and Future Work should be discussed as well. Acknowledge any limitations of the proposed algorithm and potential areas for improvement. Suggest possible extensions or future research directions to address these limitations.

Reviewer 3 Report

This work proposed an improved version of A* algorithm for path planning of spherical robot. Power consumption and 3D distance estimation are incorporated in the model. Modeling of spherical robot is provided, and the proposed algorithm is validated through several simulations. However, more details are needed in the modeling and simulation to improve the readability of the proposed work. In my view, it should be qualified for acceptance after the comments below are properly addressed:

·      Where is the 3rd motor located in Fig. 1 that controls the pendulum? Also, is the energy system itself acting as the pendulum? The CAD model shown doesn’t look like the pendulum can swing between different positions, the authors should provide a more detailed description and figure presentation to illustration how the mechanism works.

·      What is the purpose of reference 16 and 17? Is the proposed SBOT-001 a simplified replicate of reference 16?

·      More detail is need for the functions mentioned in equation 11, the current form and context does not correlate the energy calculation with the modeling discussed in section 2.2, so far it is just a general presentation and does not apply to the spherical robot specifically.

·      I would suggest the authors to conduct simulations with more complex trajectory to validate the performance of the proposed work. The current simulations are all tend to generate a relative straight trajectory, while it is of interest how the algorithm will perform if a more complicated territory is present, for example, the whole area between start and end points is out of the dynamics limits and the robot has to detour over it even if the moving direction will be away from the target point.

·      Which equation is Fig. 7 based on?

moderate revision is needed

Round 2

Reviewer 1 Report

Accept in present form

Reviewer 3 Report

Thanks for the revision, all my comments are properly addressed.

Moderate editing of English language required